# DATA FOR FREE: FEWER-SHOT ALGORITHM LEARNING WITH PARAMETRICITY DATA AUGMENTATION

**Owen Lewis**
Google (work done at MIT)
lewiso@google.com

**Katherine Hermann**
Stanford University
hermannk@stanford.edu

## ABSTRACT

We address the problem of teaching an RNN to approximate list-processing algorithms given a small number of input-output training examples. Our approach is to generalize the idea of parametricity from programming language theory to formulate a semantic property that distinguishes common algorithms from arbitrary non-algorithmic functions. This characterization leads naturally to a learned data augmentation scheme that encourages RNNs to learn algorithmic behavior and enables small-sample learning in a variety of list-processing tasks.

## 1 INTRODUCTION

Since the earliest days of neural network research, some of the most important questions about neural models have focused on their ability to capture the crispness, systematicity and compositionality that characterize symbolic computation and human cognition (Fodor & Pylyshyn, 1988; Smolensky & Legendre, 2006), and to do so with a human-like number of examples (Lake et al., 2019). While recent studies have demonstrated promising results in training recurrent neural networks (RNNs) to approximate symbolic algorithms in domains like list manipulation (Grefenstette et al., 2015; Joulin & Mikolov, 2015), binary arithmetic (Kaiser & Sutskever, 2015), graph traversal (Graves et al., 2016), and planar geometry (Vinyals et al., 2015), the question of sample efficiency remains very much open. Difficult algorithmic problems may require tens or hundreds of thousands of labelled training examples, and even simple tasks on small inputs seem to require more data than should be necessary (Lake & Baroni, 2018).

Our goal in this paper is to teach RNNs to approximate list-processing algorithms $f :: [\texttt{Int}] \to [\texttt{Int}]$ given a small collection of input-output examples, $D = \{x_i, f(x_i)\}_{i=1}^n$. Inspired by the idea of *parametricity* (Wadler, 1989) from type theory and functional programming, we hypothesize that a feature that distinguishes many algorithms from arbitrary functions is that they commute with some family of element-wise changes to their inputs. We describe a method for learning this family from the training set $D$, and show how this learned information can be used to create an augmented training set for an RNN. Our experiments show that this augmentation scheme makes it possible to approximate algorithms from small training sets, in some cases requiring only a single example per input list length.

## 2 SETUP

**RNN inductive biases.** Our data augmentation approach is motivated by the failure patterns of unaugmented training. The confusion matrix in figure 1 shows the performance of an RNN (an LSTM (Hochreiter & Schmidhuber, 1997) with 128 hidden units) trained with ten examples to copy lists of two elements. The failure mode is clear: the model acts as an interpolating lookup table: the model tends to map the regions of input space around each training input $x_i$ to the training output $f(x_i)$. This is an entirely appropriate function model for classification, but a lookup table is clearly a poor example to follow for algorithm learning. Our approach for the remainder of this paper will be to formulate a semantic property that distinguishes algorithms from lookup tables, and then use data augmentation to nudge an RNN in an algorithmic direction.

**Parametricity.** Any computable function is technically an algorithm, but we have the intuition that some functions are more "algorithm-y" than others: an algorithm is a function that "does the same thing" to any input fed to it, while a non-algorithmic function like a lookup table has different, idiosyncratically defined behavior for each possible input. Put another way, a change to the input of an algorithm should translate systematically to a change to its output. In our `copy` example, $f([1, 9]) = [1, 9]$. If we modify the input by replacing the '9' token with a '3', then making the same substitution on the output side produces the correct equation $f([1, 3]) = [1, 3]$. In an arbitrary lookup table, by contrast, $[1, 9]$ and $[1, 3]$ are simply two different table rows, and there is no reason to expect any systematic relationship between their outputs.

We can make this intuition quantitative by drawing on a family of results in type theory and programming language theory based on the idea of type parametricity, and often called "theorems for free" (Wadler, 1989). The main result is the following: Any parametrically polymorphic (for brevity, we will simply say "polymorphic" henceforth) function $f :: [a] \rightarrow [a]$, where $a$ is a type parameter[1], commutes with element-wise application of any function $g :: a \rightarrow a$:

$$f \circ \texttt{map } g = (\texttt{map } g) \circ f \qquad (1)$$

An illustrative example: doubling each of the elements of an integer list and then reversing the result gives the same output as reversing and then doubling.

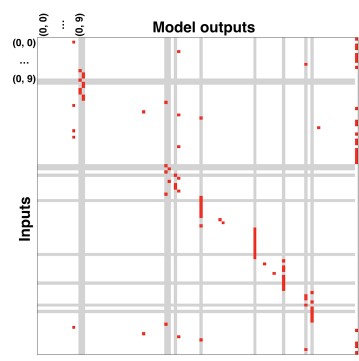

Figure 1: The confusion matrix for an RNN trained with ten examples to implement the `copy` algorithm. The inputs (rows) and outputs (columns) are each pairs of digits 0-9. A red pixel in cell $(i, j)$ indicates that the model predicted output sequence $j$ for input sequence $i$; perfect performance for this `copy` task would fill the diagonal of the heatmap with red pixels. The gray stripes indicate the ten training examples.

Parametricity captures some intuitions about what makes a function algorithmic; it accounts for the copy example above, for instance, and in general, if a function commutes with element-wise transformations of its inputs, it cannot depend in a lookup-y way on the details of its input's elements.

Since our interest is not limited to polymorphic functions, some generalization of equation 1 is required. The function `drop_evens`, which removes the even elements from a list of integers, for instance, is a clearly legitimate algorithm that fails to obey equation 1; $g : x \mapsto 2x$ is a counterexample. But while `drop_evens` does not commute with *all* element-wise transformations, it does commute with a subset of them, namely those that preserve parity. So this motivates a hypothesis: an algorithm $f :: [\texttt{Int}] \rightarrow [\texttt{Int}]$ should have some systematicity in its definition that is reflected in its commuting with some set of element-wise transformations $G = \{g :: \texttt{Int} \rightarrow \texttt{Int} \mid f \circ \texttt{map} g = \texttt{map} g \circ f\}$. We can draw a high-level analogy here with techniques from math and physics that characterize an object in terms of its symmetries.

If we can learn $G$, we can use it to augment our training set to help learn $f$: given an input-output training example $(x, f(x))$, then for any $g \in G$, $(g(x), g(f(x))$ will also be a valid training example, and can be use for data augmentation.

## 3 LEARNING COMMUTING FUNCTIONS

This section describes a technique for learning $G$ from $f$'s training data. We parameterize each $g$ as a collection of swaps $\{s_i \rightarrow t_i\}_{i=1}^m$, each of which replaces each instance of an $s$ token with the corresponding $t$ token. For instance, if $g = \{3 \rightarrow 4, 7 \rightarrow 1\}$, then $g([3, 7, 1]) = [4, 1, 1]$. Our approach to learning $g$ will be to train a classifier $C$ that predicts whether or not a given collection of swaps should commute with $f$. Given two training input-output pairs $(x_i, f(x_i))$ and $(x_j, f(x_j))$, chosen so that $x_i$ and $x_j$ have the same length, we first determine whether or not $x_i$ and $x_j$ are related by a set of swaps. Supposing there is a swap set $g$ with $g(x_i) = x_j$, we then see whether $g$ also relates the output lists $g(f(x_i)) \stackrel{?}{=} f(x_j)$: if it does, then we have a positive training example of a swap set that commutes with $f$, and if it does not, then we have a negative example.

[1]Because $f$ is parametrically polymorphic, $a$ must not be restricted to a particular type class.

Repeating this process for each pair of length-matched lists in our training set, we obtain a collection of training examples of positive (commutes with $f$) and negative (does not commute with $f$) swap sets. A promising feature of this setup is that while the original learning problem (learning $f$) has $n$ training examples, the problem of learning $G$ has $O(n^2)$ examples.

To support small-sample learning, we make the simplifying factorization assumption that each swap contributes independently to the commutativity of the whole swap set. Our classifier $C$ acts on a swap set $g = \{g_1, \ldots, g_m\} = \{s_1 \to t_1, \ldots, s_m \to t_m\}$ by $C(g) = \text{smoothed\_min}_{i=1}^m c(g_i)$, where $c$ classifies individual swaps, and $\text{smoothed\_min}(\mathbf{v}) = -\gamma \log \sum_i e^{-v_i/\gamma}$ (Cuturi & Blondel, 2017). For the experiments in this paper, $c$ consisted of a bilinear layer that combined ten-dimensional embeddings of $s_i$ and $t_i$, followed by a RELU nonlinearity and a linear layer with a scalar output.

**Per-sequence classification.** We have assumed that a given swap can be classified independently of the input list to which it is to be applied. This assumption is violated by functions like `sort`: $g = \{3 \to 6\}$ commutes with `sort` for the input list $[8, 7, 1, 3]$, but not for the list $[5, 7, 1, 3]$, for example. To deal with this, we distinguish *per-task* classification, as described above, from *per-sequence* classification, which depends on the input list elements in addition to the swap set. Extending the factorization assumption from the per-task model, for a swap set $g = \{g_1, \ldots g_m\}$ and input sequence $x = [x_1, \ldots x_k]$, the per-sequence classifier predicts $C(g, x) = \text{smoothed\_min}(\{c(g_i, x_j) \mid i = 1, \ldots m, j = 1, \ldots k\}$. Here, $c$ extends the per-task classifier by adding a bilinear combination of the sequence element with the combination of the swap elements.

**Augmentation.** To use a trained classifier $C$ to generate an augmented training set, we randomly generate a large collection of candidate swap sets $g$, recording for each the classifier score $C(g)$. Each combination of a swap set $g$ and training pair $(x, f(x))$ gives a candidate augmentation example $(g(x), g(f(x)))$. Any given candidate augmentation example could be generated in multiple ways by starting from different input-output pairs, so we assign to each candidate example the sum of the scores of the swap sets $g$ that could be used to generate it. We take our augmentation training set to be the 1000 top-scoring candidates for each list length. For both the per-sequence and per-task models, if our classifier's training set is entirely missing either positive or negative examples, the model reverts to random augmentation.

**Related work.** Learned data augmentation has been applied in computer vision, where Cubuk et al. (2018) used a reinforcement learning procedure similar to neural architecture search to learn an augmentation policy for object recognition, and NLP, where Jia & Liang (2016) used a symbolic grammar-induction procedure to learn an augmentation model for semantic parsing.

## 4 EXPERIMENTS

We compare four models: the per-task and per-sequence augmentation models described above, a random augmentation model that samples swap sets uniformly at random, and a no-augmentation baseline. We evaluate each on the list-processing functions shown in figure 2. We divide these functions into three types: polymorphic, token-dependent (depends on the identities of individual tokens), and order-dependent (depends on the order relations between tokens). These function categories are used in post-hoc analysis only; they are not explicitly made available to our models during training or testing. Expecting problem difficulty to scale to some extent with output sequence length, we also note for each function the length of its output when evaluated on an input list of length $n$. Functions that implement a filtering operation will have variable length outputs, but for simplicity, we group them with functions like `copy` in the "$\approx n$" output-length category.

For each target function, we trained our models on even-length lists of lengths 2, 4, 6 and 8, and tested on all lengths 2-8. All list items were integers in the range $[0, 19]$. To evaluate our models' sample efficiency, we considered learning problems with 1, 5, 10, 20, 30, 40, and 50 input-output examples per input list length. For all problems, our model was an LSTM with two hidden layers of 128 units.

**Results.** On the polymorphic target functions, all augmentation models, including the random one, substantially outperformed the non-augmented baseline. This performance is to be expected, as equation 1 guarantees that these functions commute with all element-wise input transformations. For functions with output length 1 and $n$, the augmented models are able to achieve close to perfect accuracy with only a single example per input list length.

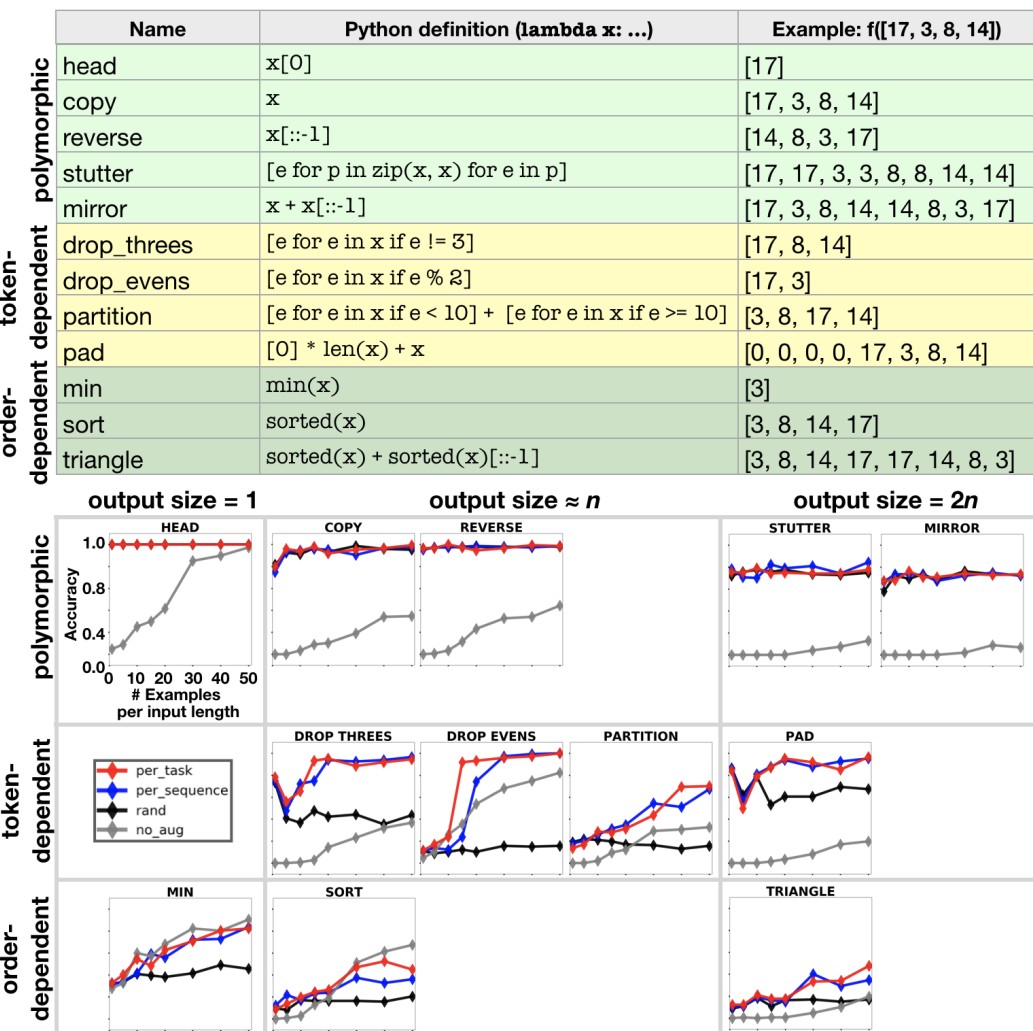

| | Name | Python definition (`lambda x: ...`) | Example: f([17, 3, 8, 14]) |
|---|---|---|---|
| head | head | `x[0]` | [17] |
| copy | copy | `x` | [17, 3, 8, 14] |
| reverse | reverse | `x[::-1]` | [14, 8, 3, 17] |
| stutter | stutter | `[e for p in zip(x, x) for e in p]` | [17, 17, 3, 3, 8, 8, 14, 14] |
| mirror | mirror | `x + x[::-1]` | [17, 3, 8, 14, 14, 8, 3, 17] |
| drop_threes | drop_threes | `[e for e in x if e != 3]` | [17, 8, 14] |
| drop_evens | drop_evens | `[e for e in x if e % 2]` | [17, 3] |
| partition | partition | `[e for e in x if e < 10] + [e for e in x if e >= 10]` | [3, 8, 17, 14] |
| pad | pad | `[0] * len(x) + x` | [0, 0, 0, 0, 17, 3, 8, 14] |
| min | min | `min(x)` | [3] |
| sort | sort | `sorted(x)` | [3, 8, 14, 17] |
| triangle | triangle | `sorted(x) + sorted(x)[::-1]` | [3, 8, 14, 17, 17, 14, 8, 3] |

Figure 2: Model performance (top) when learning the list functions (bottom) as a function of the number of training examples provided for each input list length. All accuracies reported are mean sequence accuracy (1 for a completely correct sequence, 0 otherwise).

Moving to the non-polymorphic token-dependent functions, random augmentation not only ceases to suffice for good performance, but in fact often delivers lower accuracy than the non-augmented baseline, while both learned augmentation models continue to perform well on most target functions.

For the order-dependent functions, the analysis in section 2 suggests that per-sequence augmentation should outperform per-task. In practice, the two learned models achieve roughly equal accuracy, perhaps reflecting the fact that the more expressive per-sequence model requires more data to train correctly. Moreover, while the learned augmentation models outperform the non-augmented baseline on the `triangle` task and on the very few-shot versions of the sorting tasks, their advantage over the non-augmented baseline is much less marked than on the other function types. Still, the learned models largely avoid the destructive effects of applying random augmentation.

**Future directions.** Our augmentation schemes are model-agnostic; it will be interesting in the future to pair them with models like pointer networks or memory-augmented RNNs. It will also be interesting to extend the techniques of this paper to domains beyond list processing, for instance to the geometric algorithms studied in (Vinyals et al., 2015).

ACKNOWLEDGMENTS

The first author was supported by the Center for Brains, Minds and Machines (CBMM), funded by NSF STC award CCF-1231216.

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
