# OpenReview forum: "Data for free: Fewer-shot algorithm learning with parametricity data augmentation"
_ICLR.cc/2019/Workshop/LLD — LLD 2019_

### Official Review · AnonReviewer2 · 2019-04-07
**Interesting Paper**

**Rating:** 5
**Confidence:** 3

**Review:**


Notes:
  -Problem of teaching an RNN to approximate list processing algorithms from a small number of examples.

  -Learned data augmentation scheme based on parametricity from PL theory.

  -Helps with small-sample learning.

  -Algorithmic problems seem to require more training data than should be necessary.

  -List processing algorithms map from an input list to an output list.

  -Many algorithms are distinguished from arbitrary functions is that they commute over elementwise changes to the inputs.

  -Commutative means applying a function to each element of the list and running the list processor

  -While I like the basic idea that there are logical properties that algorithms should obey, the commutative property for arbitrary functions seems too strong to me.  To give one example: multiplying by -1 does not commute with list sorting.  (note: the paper addresses this later).

Summary: This is a very good workshop paper which presents a simple idea

Some ideas for future work and directions:

  -It might be interesting to consider enforcing this type of commutative property in the hidden states.  It in some ways would require reversing your way of thinking - because you'd need to think about what properties the hidden states would need to have to allow the sequential part to be commutative, but it would have a big advantage that it would require a less data-dependent way of deciding if the function commutes or not.

  -There is a technique called Mixup (Zhang 2018) as well as Interpolation Consistency Training (Verma 2019) which tries to encourage linear combination of input examples to map to the corresponding interpolations of the outputs.  If you write the interpolation as a function x, you can rewrite this as: mix(f(x)) = f(mix(x)), where f is the neural network (this is literally what (Verma 2019) enforces and (Zhang 2018) does something slightly different).  Thus it is encouraging interpolations to commute with the neural network function.

---

### Official Review · AnonReviewer1 · 2019-04-14

**Rating:** 4
**Confidence:** 1

**Review:**

The paper tackles the problem of teaching an RNNs to approximate list-processing algorithms. The authors argue that what distinguishes algorithms from arbitrary functions is that they commute with a family of element-wise changes to their inputs and propose a method that learns RNN functions to approximate such family from data.

To learn commuting functions, the authors propose to synthetically generate labeled data by testing whether a function commutes with a collection of swaps. The corresponding classifier that approximates commutative swap functions is used for data augmentation. I find the observation about the parametricity property from type theory is interesting and the proposed data augmentation approach seems novel and interesting.

The only concern I have (perhaps, stemming from a mild misunderstanding of the method), if the proposed approach would work beyond the simple inputs of integer sequences (i.e., with more complex input-output data types, such as images, text, sounds, etc. as most of the modern machine learning has to deal with), or there are potential limitations that need to be resolved.

---

### Decision · Program_Chairs · 2019-04-15
**Acceptance Decision**

Accept